# Characterization of Prophages and Their Genetic Cargo in Clinical *M. abscessus* Isolates

**DOI:** 10.3390/microorganisms13092028

**Published:** 2025-08-30

**Authors:** Sichun Luan, Yu Chen, Na Li, Qing Miao, Jue Pan, Bijie Hu

**Affiliations:** Department of Infectious Diseases, Zhongshan Hospital, Shanghai 200032, China; luan.sichun@zs-hospital.sh.cn (S.L.);

**Keywords:** *Mycobacterium abscessus*, prophage, genome sequencing

## Abstract

Limited data exist regarding lysogenic phages carried by *M. abscessus*, as well as regarding their roles played in diseases. Strains identified as *M. abscessus* from patients were collected. Prophages, virulence factors, and antibiotic resistance genes present in genomes were predicted, and correlations between prophages, virulence factors, antibiotic resistance genes, and clinical patient prognoses were analyzed. A total of 145 prophage sequences were detected in 56 *M. abscessus* strains. Prophages contained more virulence factors and antibiotic resistance genes, compared to known mycobacteriophages. The average sequence similarity among prophage sequences from a single patient was significantly higher than that among prophages from different patients or between prophages and known phages. The study showed that *M. abscessus* commonly carries prophages, which are enriched in virulence factors and antibiotic resistance genes relative to known phages, but their relationship to clinical prognoses requires further study. Prophages present in strains from different patients were highly diverse and exhibited low similarity with known mycobacterial phages.

## 1. Introduction

*Mycobacterium abscessus* was first identified in 1953 from subcutaneous abscess-like lesions in the gluteal region following knee trauma [1,2,3]. Currently recognized as the second most prevalent pathogenic non-tuberculous mycobacteria (NTM) after M. avium [2,3], this organism predominantly colonizes patients with pre-existing pulmonary conditions, including cystic fibrosis, chronic obstructive pulmonary disease, and bronchiectasis. Knowledge of virulence determinants in NTM is limited. Genomic studies have revealed that *M. abscessus* harbors horizontally acquired *mgtC* and *plc* genes from other bacterial species, though their precise in vivo functions remain uncharacterized [3].

*M. abscessus* exhibits two distinct colony morphotypes on solid media: smooth colonies are associated with abundant glycopeptidolipids and demonstrate reduced virulence, while rough colonies are deficient in glycopeptidolipids but exhibit enhanced virulence potential. A critical virulence mechanism in pathogenic mycobacteria is the type VII secretion system (ESX). *M. abscessus* specifically maintains ESX-3 and ESX-4 systems, which facilitate intracellular survival following macrophage phagocytosis by disrupting phagosomal membranes [3,4,5].

Although not the most prevalent NTM pathogen, *M. abscessus* presents exceptional therapeutic challenges [3,5]. Its intrinsic resistance mechanisms include *arrMab*-mediated rifampin inactivation, *blaMab*-dependent degradation of imipenem and cefoxitin, and erm41 upregulation upon macrolide exposure, conferring inducible resistance [3,5]. Additionally, acquired resistance can occur via 16S rRNA and 23S rRNA mutations, potentially leading to intractable infections [3,6].

Bacteriophages represent a diverse group of viruses exhibiting strict host specificity for bacteria, fungi, actinomycetes, and spirochetes. While lytic phages mediate host cell lysis during their life cycle, lysogenic phages integrate their genomes into bacterial chromosomes for vertical transmission. Lysogenic phages frequently encode genes that modify bacterial phenotypes, including alterations in host genotype and conferring resistance to other phages of the same type. These elements significantly influence host bacterial characteristics and drug susceptibility profiles.

Emerging clinical evidence supports the application of phage therapy for treating *M. abscessus* infections, with several studies reporting favorable outcomes [6,7]. Several genomic investigations have revealed complex phage–mycobacteria interactions. Glickman et al. documented widespread prophage distribution in rapidly-growing mycobacteria, with an enrichment of virulence factors that exceeded that found in characterized phages [8]. Dedrick et al. isolated nine unique lysogenic phages from *M. abscessus*, demonstrating distinct host ranges compared to established mycobacteriophages [9].

This study performed computational prophage prediction and analysis of virulence factors and antibiotic resistance genes in prophages of clinical *M. abscessus* strains, investigating relationships between prophage sequences, genes encoded by prophages, and clinical manifestations to comprehend potential integrated prophages in clinical *M. abscessus* more clearly.

## 2. Materials and Methods

### 2.1. Culturing Clinical Strains

Clinical strains identified as *M. abscessus* by MALDI-TOF-MS isolated from September 2017 to August 2021 and preserved in the clinical microbiology laboratory were collected and revived. Multiple strains isolated from the same patient were excluded unless isolated at intervals exceeding one month. Clinical manifestations, diagnoses and prognoses, and antibiotic susceptibility test results were collected. Single-colony strains were cultured, inoculated, and isolated using 7H10 culture medium supplemented with Oleic Albumin Dextrose Catalase. Genome sequencing was performed by Shanghai Majorbio Inc. (Shanghai, China).

### 2.2. Genome Sequencing, Annotation, and Analysis

Further genome sequencing, annotation, and analysis were performed using the cloud bioinformatic platform provided by Shanghai Majorbio Inc. [10]. Strain subspecies were determined by comparing sequences to NCBI databases using *rpoB* sequences, and a minimum-evolution tree was constructed based on *rpoB* sequence differences. Virulence factors and resistance genes were analyzed using the cloud platform.

### 2.3. Prophage Sequence Prediction

Prophage sequence prediction was performed using the PHASTER (https://phaster.ca, accessed on 1 October 2022) online tool [10,11,12]. PHASTER scans input genome sequences and output prophage sequences, the total number of genes, names of similar phages, and a gene annotation map. PHASTER also assesses whether a sequence is intact and assigns one of three intactness levels (Intact, Questionable, or Incomplete) to the predicted sequence according to a specific algorithm described previously [10,11,12].

### 2.4. Prophage Gene Annotation and Library Construction

A virulence factor database and an antibiotic resistance gene database were downloaded from VFDB and CARD databases and served as target libraries [13,14]. Based on known similar phages identified by PHASTER, a known phage database was downloaded from PhagesDB and constructed. The Diamond tool (Ver. 0.9.14) was used to compare genes within the prophage library and the known phage library against the VFDB and CARD databases to annotate virulence factors and resistance genes, enabling the determination of virulence factor and resistance gene proportions and their associations with prophage sequences.

### 2.5. Prophage Sequence Alignment

The BLAST+ tool (Ver. 2.13.0) was used to perform pairwise alignment within the prophage sequence library to determine sequence differences. Alignments were divided into 2 categories: alignments among strains from the same patient and alignments between strains from different patients. Alignments among strains from the same patient primarily focused on sequence differences, and alignments between strains from different patients focused on phage sequence similarities and shared genes.

### 2.6. Prophage Sequences and Patient Prognoses

Patients were divided into 2 groups according to treatment prognoses, and potential relationships between prophage gene content and prognoses were analyzed. Shared genes across prophages were also analyzed to investigate potential relationships with prognoses. Unfavorable outcomes included treatment failure, treatment discontinuation, and loss to follow-up; and favorable outcomes included: microbiological conversion and clinical cure without microbiological conversion, all defined according to the China Clinical Diagnosis and Treatment Guidelines for NTM-PD (2020 Edition) [15].

### 2.7. Patient Informed Consent and Ethical Approval

For all patients enrolled in this retrospective study, informed consent forms for biological sample donation were signed upon admission. Patients who did not sign informed consents or refused to donate samples upon admission were excluded from this study. This retrospective study was approved by the Ethics Committee of Zhongshan Hospital, Fudan University, under approval No. B2022-513R. Regarding bacterial genome data availability, all data have been deposited in OMIX at the China National Center for Bioinformation/Beijing Institute of Genomics, Chinese Academy of Sciences (https://ngdc.cncb.ac.cn/omix, accessed on 1 October 2025, accession No. OMIX010879). Generative artificial intelligence (GenAI) was not used to generate any text, data, or graphics or to assist in study design, data collection, analysis, or interpretation.

### 2.8. Statistical Analysis Methods

One-way ANOVA was used to analyze prophage sequence characteristics and encoded gene differences. The Wilcoxon rank-sum test was used to analyze differences in gene numbers, gene proportions, and phage characteristics between patients with different prognoses. Statistical analysis was performed using IBM SPSS Statistics 26.0. Graphs were generated by GraphPad Prism 9.5.0. Sequence alignments were performed using NCBI Blast+ 2.13.0 and Diamond 0.9.14.

## 3. Results

### 3.1. Patient Characteristics and Prognoses

Initially, 69 strain samples preserved in the clinical microbiology laboratory from September 2017 to August 2021 were collected. Ultimately, the study included 26 patients with *M. abscessus* infection and a total of 56 strains identified as *M. abscessus*, as 11 samples were not analyzable due to DNA degradation, and 2 samples were excluded due to contamination. A total of 16 patients yielded only a single strain, while 10 patients yielded two or more strains. According to the China Clinical Diagnosis and Treatment Guidelines for NTM-PD (2020 Edition) [15], patient outcomes were evaluated by reviewing hospital inpatient records, outpatient visit records, and laboratory test results up to December 2022. Excluding 1 patient with vertebral NTM infection for whom prognosis could not be determined, the outcomes were: treatment failure in 14 patients (53.8%), treatment discontinuation in 1 patient (3.8%), loss to follow-up in 4 patients (15.4%), microbiological conversion in 1 patient (3.8%), and clinical cure without microbiological conversion in 5 patients (19.2%) (Figure 1).

### 3.2. Genome Analysis and Prophage Prediction of M. abscessus

Based on BLAST comparisons of bacterial *rpoB* sequences with the NCBI database, 51 strains were identified as *M. abscessus* subsp. *abscessus*, 3 as *M. abscessus* subsp. *massiliense*, and 2 as *M. abscessus* subsp. *bolletii*. Whole-genome analysis of the 56 strains detected an average of 354 ± 28.7 virulence factors (95% CI: 346.4–361.6) and 152.9 ± 18.1 resistance genes (95% CI: 148.1–157.7) per strain. The phylogenetic tree of the *M. abscessus* strains and detailed sequencing parameters are provided in Appendix A.

PHASTER analysis of the 56 strains identified a total of 145 prophage sequences, categorized as: 37 intact (25.5%), 17 questionable (11.7%), and 91 incomplete (62.8%). All clinical strains contained at least one prophage sequence (100%, 56/56), and 32 strains (57.1%, 32/56) harbored at least one intact prophage. The average numbers of intact, questionable, and incomplete prophage sequences per strain were 0.7 ± 0.6 (95% CI: 0.5–0.8), 0.3 ± 0.6 (95% CI: 0.2–0.5), and 1.6 ± 1.2 (95% CI: 1.3–1.9), respectively. The distribution and number of prophages per bacterial genome are detailed in Figure 2a. One strain initially identified as *Mycobacterium intracellulare* and excluded from the study was also analyzed for prophages, but no prophage sequence was detected.

The prophage sequences detected in this study had an average length of 26,174 ± 17,488 bp (95% CI: 23,314–29,035) and encoded an average of 33.5 ± 24.1 proteins (95% CI: 29.6–37.5). Of these, 24.5 ± 16.9 (95% CI: 21.7–27.2) proteins were confirmed as phage proteins, and 9.0 ± 8.3 were predicted as hypothetical proteins. Intact prophages had an average length of 47,224 ± 14,967 bp (95% CI: 42,160–52,288), while questionable prophages averaged 27,069 ± 7967 bp (95% CI: 23,107–31,031), and incomplete prophages averaged 16,558 ± 8770 bp (95% CI: 14,721–18,394). Intact prophages encoded an average of 66.1 ± 20.3 genes (95% CI: 59.4–72.9), questionable prophages encoded 35.4 ± 7.8 genes (95% CI: 31.5–39.3), and incomplete prophages encoded 19.6 ± 11.2 genes (95% CI: 17.3–22.0) (Figure 2b,c). One-way ANOVA revealed statistically significant differences in sequence length and gene counts among the three prophage categories (*p* < 0.0001). Post hoc pairwise comparisons showed statistically significant differences for all comparisons between categories.

### 3.3. Analysis of Virulence Factors and Resistance Genes in Prophages and Known Phages

Whole-genome analysis revealed that the 56 *M. abscessus* strains contained an average of 354.0 ± 28.7 virulence factors per strain (95% CI: 346.4–361.6), predominantly including adherence factors (16.5%), iron uptake systems (28.5%), and secretion systems (12.6%). Resistance gene analysis identified an average of 152.9 ± 18.1 genes per strain (95% CI: 148.1–157.7), targeting rifampicin (12.8%), fluoroquinolones (13.9%), carbapenems (14.0%), macrolides (19.8%), and tetracyclines (25.9%) as the most common classes (Figure 3a,b). Detailed data are shown in Appendix A.

Prophage sequences contained a total of 114 virulence factors and 43 resistance genes, accounting for 2.3% and 0.9% of total prophage-encoded proteins, respectively. Among these, 35 virulence factors were mycobacterial-specific (e.g., type VII secretion system genes, *whiB3*), while others had origins in other bacteria. Resistance genes included efflux pumps (*Abau_AbaF*), β-lactamases (PDC-364), and nitroimidazole resistance genes (*msbA*) (Figure 3c,d). Comparison with 53 known phages predicted to be similar by PHASTER revealed they contained thirty-six virulence factors and five resistance genes (Figure 3e,f). Wilcoxon tests showed that prophages carried more resistance genes (*p* = 0.0248) and had higher proportions of virulence factors (*p* = 0.0280) and resistance genes (*p* = 0.0264) than known phages. Detailed data are shown in Appendix A.

### 3.4. Comparative Analysis of Virulence Factors and Resistance Genes

On average, each prophage sequence contained 0.8 ± 0.9 (95% CI: 0.6–0.9) virulence factors and 0.3 ± 0.8 (95% CI: 0.2–0.4) resistance genes per sequence. In contrast, each known phage contained an average of 0.7 ± 0.8 (95% CI: 0.5–0.9) virulence factors and 0.1 ± 0.5 (95% CI: 0.0–0.2) resistance genes. Wilcoxon rank-sum tests revealed no significant difference in the number of virulence factors per phage/prophage (*p* = 0.829), but prophages carried significantly more resistance genes per sequence (*p* = 0.0248). Virulence factors and resistance genes accounted for 4.4 ± 7.4% (95% CI: 3.2–5.6%) and 2.6 ± 9.8% (95% CI: 1.0–4.2%) of prophage-encoded proteins, respectively, compared to 0.7 ± 0.6% (95% CI: 0.5–1.0%) and 0.1 ± 0.5% (95% CI: 0.0–0.3%) in known phages. Prophages exhibited significantly higher proportions of both gene types (*p* = 0.0280 for virulence factors; *p* = 0.0264 for resistance genes) (Figure 4a–d).

The whole bacterial genomes contained 354.0 ± 28.7 virulence factors per genome (95% CI: 346.4–361.6) and 152.9 ± 18.1 resistance genes per genome (95% CI: 148.1–157.7), while the prophages within each genome averaged 2.0 ± 1.7 virulence factors (95% CI: 1.6–2.5) and 0.8 ± 1.1 resistance genes (95% CI: 0.5–1.1) per genome. Virulence factors and resistance genes constituted 6.9 ± 0.3% (95% CI: 6.8–7.0%) and 3.0 ± 0.1% (95% CI: 3.0–3.0%) of all bacterial proteins, respectively, versus 5.1 ± 7.9% (95% CI: 3.0–7.2%) and 3.0 ± 6.2% (95% CI: 1.3–4.6%) of proteins encoded in prophages. Wilcoxon paired tests confirmed significant differences in these gene proportions (*p* = 0.0086 for virulence factors; *p* = 0.0066 for resistance genes) (Figure 4e,f).

### 3.5. Association Between Bacterial Resistance Genes and Clinical Manifestations

Antimicrobial susceptibility testing (AST) results for 28 strains revealed high resistance to tetracyclines (e.g., 96.4% resistance to doxycycline), fluoroquinolones (71.4–78.6% resistance), and sulfonamides (96.4% resistance to SMZ/TMP). In contrast, all strains were sensitive to amikacin (100%), while intermediate susceptibility predominated for tobramycin (57.1%), cefoxitin (67.9%), and imipenem (60.7%). Regarding clarithromycin, 26 strains (92.3%) were sensitive on day 5 (D5), but only 7 strains (25.0%) remained sensitive on day 14 (D14), indicating a high rate of induced resistance to macrolides. No significant correlations were observed between the number of resistance genes and AST profiles, except for tobramycin (*p* = 0.0316), though this finding lacked clear clinical relevance (Appendix A).

The 26 patients with determined prognoses were categorized based on clinical outcomes to compare the mean values of the number of prophages per strain, number of phage-encoded proteins per strain, number of virulence factors encoded within prophages per strain, number of resistance genes encoded within prophages per strain, and their proportions relative to the total prophage-encoded proteins per strain.

In the favorable prognosis group, isolates harbored an average of 2.8 ± 1.8 prophages per strain (95% CI: 0.9–4.8), encoding 90.8 ± 48.6 proteins in prophages per strain (95% CI: 39.8–141.8). Prophage sequences contained 3.7 ± 2.2 virulence factors per strain (95% CI: 1.4–5.9) and 0.5 ± 1.2 resistance genes per strain (95% CI: −0.8–1.8), accounting for 6.84 ± 7.45% (95% CI: −0.98–14.65%) and 3.33 ± 8.17% (95% CI: −5.23–11.90%) of the prophage-encoded proteins per strain, respectively.

In the unfavorable prognosis group, isolates carried 2.5 ± 1.3 prophages per strain (95% CI: 2.1–2.9), encoding 87.9 ± 66.0 proteins in prophages per strain (95% CI: 68.5–107.2). Prophages contained 1.8 ± 1.5 virulence factors per strain (95% CI: 1.3–2.2) and 0.8 ± 1.1 resistance genes per strain (95% CI: 0.5–1.1), representing 4.94 ± 8.26% (95% CI: 2.51–7.36%) and 2.98 ± 6.21% (95% CI: 1.16–4.81%) of the encoded proteins per strain, respectively.

Wilcoxon rank-sum tests revealed that only the difference in the number of virulence factors encoded within prophages per strain reached statistical significance (*p* < 0.05). No significant differences were observed in the number of prophages per strain, number of proteins encoded in prophages per strain, number of resistance genes encoded within prophages per strain, or the gene proportions relative to total prophage-encoded proteins. Notably, strains from patients with favorable outcomes carried significantly more prophage-encoded virulence factors per strain than those with poor prognoses (Figure 5).

### 3.6. Comparative Analysis of Prophage Sequence Homology

After excluding self-matches, sequence alignments among strains from the same patients yielded 789 hits with an average sequence identity rate of 99.86 ± 0.41% (95% CI: 99.83–99.89%). The mean aligned sequence length was 14,814 ± 13,546 bp (95% CI: 13,867–15,760 bp), with an average of only 0.6 ± 1.2 bp mismatches (95% CI: 0.5–0.6 bp) per aligned sequence. Alignments between strains from different patients produced 4503 hits showing significantly lower similarity: average sequence identity rate 89.27 ± 7.56% (95% CI: 89.05–89.50%), mean aligned length 2100 ± 3577 bp (95% CI: 1995–2204 bp), and 122.0 ± 132.6 bp mismatches (95% CI: 118.2–125.9 bp). Alignment with known mycobacteriophages resulted in 355 hits with further reduced similarity: sequence identity rate 78.28 ± 6.98% (95% CI: 77.55–79.01%), mean aligned length 1193 ± 2415 bp (95% CI: 941.2–1445 bp), and 210.2 ± 158.6 bp mismatches (95% CI: 193.7–226.8 bp). Kruskal–Wallis tests demonstrated statistically significant differences (*p* < 0.0001 for all comparisons) in both sequence identity rates and mismatch frequencies among intra-patient, inter-patient, and known phage alignment groups (Figure 6).

Notably, strains from patients PAT08, PAT14, and PAT17 shared two incomplete prophages (9336 bp and 6301 bp) with identical sequences, which was consistent with their clustering on the same minor phylogenetic branch. This might suggest that these three patients were infected with potentially similar strains, though this requires further investigation. These prophages also showed partial homology with strains from PAT10, PAT19, and PAT23.

### 3.7. Association Between Common Genes in Prophages and Patient Prognosis

Analysis of prophage-encoded virulence factors and resistance genes revealed several conserved elements. Among resistance genes, PDC-364, *Abau_AbaF, novA,* and *msbA* were most prevalent, each detected in ≥8 prophages, collectively accounting for 90.5% (38/42) of all identified resistance genes within prophages. The predominant virulence factors included *clpP, fleN, GroEL, vapA1, whiB3, irtA,* and *atf*, representing 61.4% (70/114) of total virulence factor detections within prophages, with *clpP, fleN, GroEL*, and *vapA1* each detected in >10 prophages (Figure 7).

To evaluate potential prognostic implications, we analyzed the distribution of the seven most frequent virulence factors (*clpP, fleN, GroEL, vapA1, whiB3, irtA,* and *atf*) and four most frequent resistance genes (PDC-364, *Abau_AbaF, novA,* and *msbA*) encoded within prophages across clinical isolates from 25 patients with documented outcomes. The genes *fleN* and *GroEL* showed the highest patient prevalence (detected in prophages from n = 8 patients each), while *vapA1* was least frequent (detected in prophages from n = 2 patients). Detailed distributions are presented in Table 1.

Chi-square testing revealed no significant positive correlations between most tested genes and clinical outcomes. However, *whiB3* demonstrated statistically significant enrichment in patients with favorable prognoses (*p* < 0.05), suggesting its potential role as a prognostic marker. This observation warrants further investigation to elucidate the functional significance of prophage-encoded *whiB3* in *M. abscessus* infections.

## 4. Discussion

This study analyzed clinical isolates of *M. abscessus* collected from September 2017 to August 2021 at our hospital’s clinical microbiology laboratory. Following species identification, we performed whole-genome sequencing, annotation, and prophage analysis to investigate the distribution of bacterial genome- and prophage-encoded virulence factors and resistance genes; genomic differences between *M. abscessus* prophages and known mycobacteriophages; and potential associations with clinical outcomes. The sample size was limited and uneven due to restricted patient availability and treatment challenges during the study period; further studies with larger cohorts are warranted.

### 4.1. Genomic Sequencing, Composition Analysis, and Phylogenetic Relationships

We conducted deep-coverage sequencing of 56 *M. abscessus* isolates. Consistent with Jin P. et al.’s 2022 study [16] of 69 clinical strains (2014–2016), phylogenetic analysis revealed three major clades corresponding to *M. abscessus* subsp. *abscessus*, *massiliense*, and *bolletii*. All serial isolates from individual patients clustered within identical clades, demonstrating strain stability over extended infection periods (up to 29 months between isolates). While inter-patient strains showed phylogenetic similarity, they maintained high genetic diversity. The observed phylogenetic distances between subspecies align with established taxonomic classifications. The absolute majority isolates were *M. abscessus* subsp. *abscessus* strains from NTM-PD, so it was hard to tell whether lineage differences would confound the result or not.

### 4.2. Prophage Content Analysis in M. abscessus

Among 56 analyzed strains, we identified 145 prophage sequences in the *M. abscessus* genomes. Notably, one *M. intracellulare* strain excluded during species identification lacked detectable prophages. Current research on non-tuberculous mycobacterial prophages remains limited. Glickman et al. (2020) [8] reported intact prophages in 56.0% of rapidly growing mycobacteria (RGM, primarily *M. abscessus* complex) versus only 6.19% of slow-growing mycobacteria (SGM, e.g., *M. avium, M. intracellulare*), suggesting RGM-specific prophage enrichment. Dedrick et al. (2021) [9] similarly observed abundant prophages in *M. abscessus*. Our findings corroborate these reports, with 100% of *M. abscessus* strains harboring prophages (57.1% containing intact sequences), contrasting with the lack of prophages in the *M. intracellulare* strain. Since repeated strains from the same patient varied from prophages, virulence factors, and resistance genes detected, the study conserved these strains, and further studies with larger cohorts are warranted.

Technical limitations of next-generation sequencing may underestimate prophage detection, as Glickman et al. [8] noted a higher frequency of contig-edge prophages in RGM, potentially due to DNA fragmentation; nevertheless, RGM consistently showed greater prophage content than SGM. While growth rate differences were hypothesized as a potential explanation [8], the mechanistic basis for this enrichment remains unclear and warrants further investigation.

### 4.3. Genomic Analysis of Virulence Factors and Resistance Genes in M. abscessus Prophages

Compared to known mycobacteriophages, prophages contained increased numbers of resistance genes per sequence and higher proportions of both virulence factors and resistance genes among their encoded proteins. However, whole bacterial genomes contained greater absolute numbers of these genes, suggesting prophages contribute to but are not the primary source of these genetic elements. Glickman et al. [8] similarly observed virulence factor enrichment in clinical prophages (1.01%) versus environmental phages (0.13%).

Virulence factor profiles differed between bacterial genomes (dominated by adherence factors, iron acquisition systems, and type VII secretion systems) and prophages (enriched for adaptation genes, type VII secretion systems, and stress proteins). Resistance genes in prophages included efflux pumps, β-lactamases, and genes conferring macrolide resistance genes, partially overlapping with chromosomal resistance mechanisms reported by Jin et al. [16] and Dedrick et al. [9]. This study further revealed prophages as potential untapped sources of virulence and resistance genes in *M. abscessus*.

### 4.4. Clinical Correlation of Resistance Genes and Antimicrobial Susceptibility

Retrospective analysis of 28 strains with available antimicrobial susceptibility testing (AST) data revealed no significant correlations between the number of resistance genes and phenotypic resistance profiles. This aligns with Johansen et al.’s [3] review of *M. abscessus* resistance mechanisms, including innate resistance via *blaMab*-mediated β-lactam degradation and *MabTetX*-mediated tetracycline inactivation; inducible macrolide resistance through *erm41* upregulation; and acquired resistance via rRNA mutations. The study was unable to evaluate subspecies-specific resistance patterns due to limited AST data for certain subgroups.

### 4.5. Clinical Relevance of Prophages in Patient Outcomes

No significant associations emerged between prophage quantity/completeness or virulence factor/resistance gene content within strains with clinical prognosis. Paradoxically, favorable outcomes correlated with strains carrying higher numbers of prophage-encoded virulence factors per strain. As the functional activity of these genetic elements was unconfirmed, their clinical impact requires validation through transcriptional and proteomic studies. Dedrick et al. [9] successfully reactivated functional phages from prophages in *M. abscessus*, observing extensive genomic rearrangements during excision—a phenomenon potentially explaining the high proportion of fragmented prophages in our sequencing data.

### 4.6. Evolutionary Dynamics of M. abscessus Prophages

Comparative sequence analysis revealed near-identical prophages (99.86 ± 0.41% sequence identity) within intra-patient serial isolates; reduced homology (89.27 ± 7.56% sequence identity) between inter-patient isolates; and minimal similarity (78.28 ± 6.98% sequence identity) compared to known phages. Kruskal–Wallis tests confirmed significant differences (*p* < 0.0001) across all comparisons. Notably, strains from patients PAT08, PAT14, and PAT17 shared two incomplete prophages (9336 bp and 6301 bp) with 100% identity, suggesting possible transmission of closely related strains. These prophages showed partial homology with strains from three additional patients. The mosaic architecture of prophages likely reflects genetic recombination and degradation [8]. Dedrick et al. [9,17] characterized *M. abscessus* phages with unique properties: divergent sequences from known mycobacteriophages; expanded host ranges (including smooth colony variants); and novel virulence factors potentially secreted via type VII systems. These findings position *M. abscessus* prophages as both potential virulence determinants and untapped resources for phage therapy development.

### 4.7. Conserved Virulence and Resistance Elements in Prophages

Analysis of prevalent virulence factors and resistance genes encoded within prophages revealed only *whiB3* correlated with prognosis (higher prevalence in favorable outcomes, *p* < 0.05). The *whiB3* regulator, known in *M. tuberculosis* to maintain redox homeostasis and facilitate granuloma formation [18,19,20], may function differently in *M. abscessus*, possibly enhancing virulence by promoting macrophage escape rather than intracellular persistence. Supporting prophage-mediated virulence enhancement, Fan et al. (2022) [21] demonstrated that the prophage gene Rv2650c from *M. tuberculosis* altered host cytokine responses (TNF-α, IL-10, IL-1β, and IL-6) and increased stress resistance in *M. smegmatis*. This underscores the need to functionally characterize *M. abscessus* prophage elements in future studies.

## 5. Conclusions

The *M. abscessus* genome extensively harbors integrated prophage sequences carrying both virulence factors and resistance genes. Compared to known environmental mycobacteriophages, *M. abscessus* prophages contain increased numbers of resistance genes per sequence and exhibit significant enrichment in the proportions of both virulence factors and resistance genes among their encoded proteins. Considerable prophage diversity exists among strains from different patients, exhibiting low similarity to known mycobacteriophages. Notably, the prophage-encoded *whiB3* gene was detected at a higher frequency in patients with favorable prognoses, though its precise mechanistic role requires further investigation. These findings highlight the complex interplay between prophage-encoded elements and *M. abscessus* pathogenesis while underscoring the need for functional studies to elucidate their clinical significance.

## Figures and Tables

**Figure 1 microorganisms-13-02028-f001:**
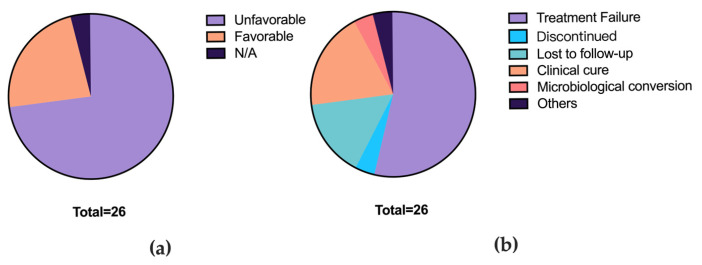
Treatment outcomes of patients with *M. abscessus* infection. (**a**) Patient divided by prognosis outcomes. (**b**) Detailed outcomes.

**Figure 2 microorganisms-13-02028-f002:**
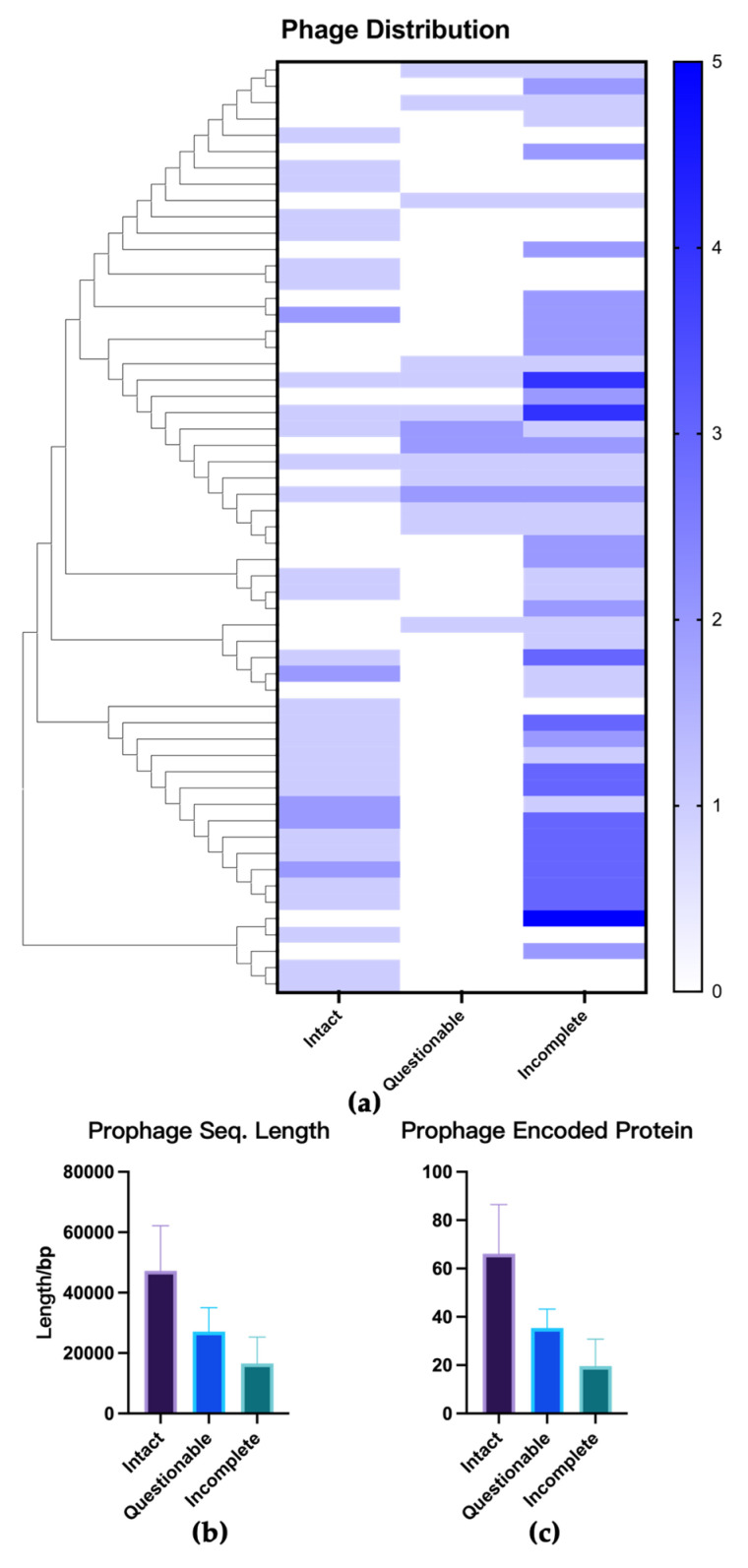
Distributions, average length, and gene counts of *M. abscessus* prophages. (**a**) Distribution of prophage sequence quantities. (**b**) Prophage average sequence length. (**c**) Prophage average encoded protein.

**Figure 3 microorganisms-13-02028-f003:**
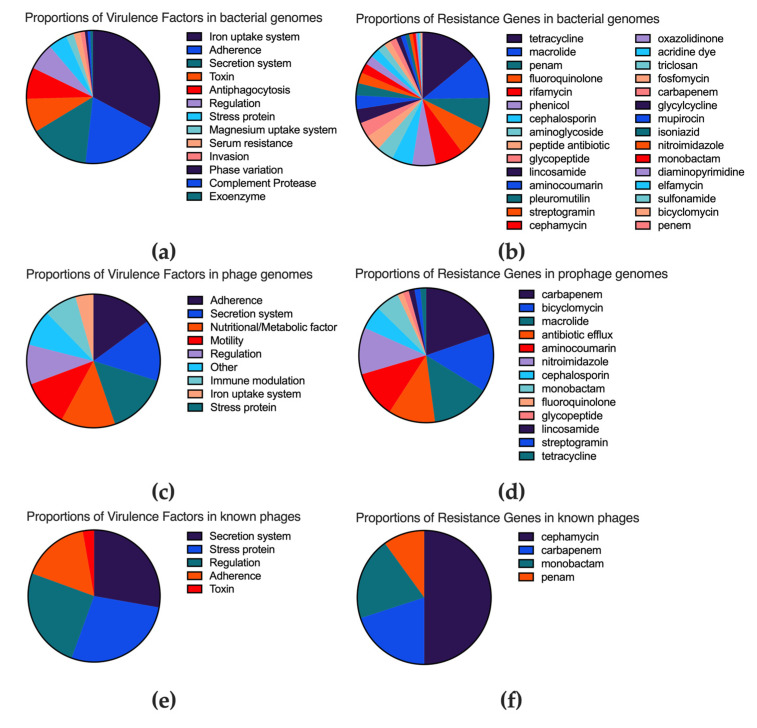
Proportions of virulence factors and resistance genes across different genomic contexts. (**a**) Proportion of virulence factor types within the entire genomes of *M. abscessus* strains. (**b**) Proportion of resistance gene types targeting different antibiotic classes within the entire genomes of *M. abscessus* strains. (**c**) Proportion of virulence factor types encoded within prophage sequences. (**d**) Proportion of resistance gene types targeting different antibiotic classes encoded within prophage sequences. (**e**) Proportion of virulence factor types encoded within known mycobacteriophage genomes. (**f**) Proportion of resistance gene types targeting different antibiotic classes encoded within known mycobacteriophage genomes.

**Figure 4 microorganisms-13-02028-f004:**
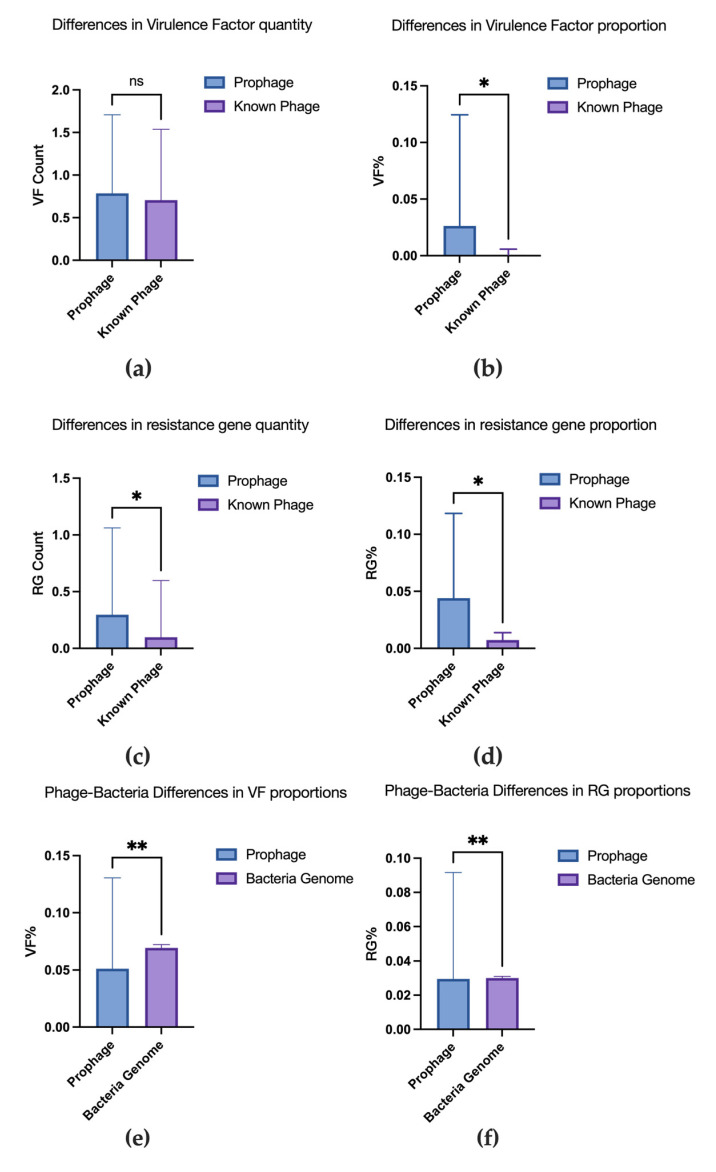
Differences in virulence factor and resistance gene quantities and proportions between prophages and known phages and between prophages and bacterial genomes. (**a**) Differences in virulence factor quantity. (**b**) Differences in virulence factor proportion. (**c**) Differences in resistance gene quantity. (**d**) Differences in resistance gene proportion. (**e**) Differences in virulence factors proportion versus bacterial genomes. (**f**) Differences in resistance gene proportion versus bacterial genomes. *: 0.01 ≤ *p* < 0.05, **: 0.001 ≤ *p* < 0.01, ns: No Significance, *p* > 0.05.

**Figure 5 microorganisms-13-02028-f005:**
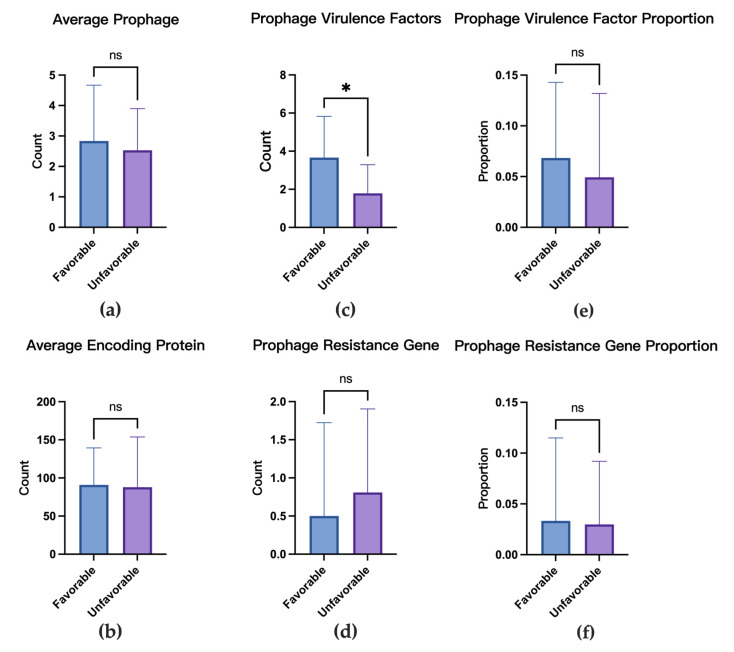
Analysis of prophage-associated metrics grouped by patient prognosis. Favorable prognosis (n = 6 patients); unfavorable prognosis (n = 19 patients). (**a**) Prophage differences between strains from different prognoses. (**b**) Prophage encoding protein differences between strains from different prognoses. (**c**) Prophage virulence factor differences. (**d**) Prophage resistance gene differences. (**e**) Prophage virulence factor proportion differences. (**f**) Prophage resistance gene proportion differences. *: 0.01 ≤ *p* < 0.05, ns: No Significance, *p* > 0.05.

**Figure 6 microorganisms-13-02028-f006:**
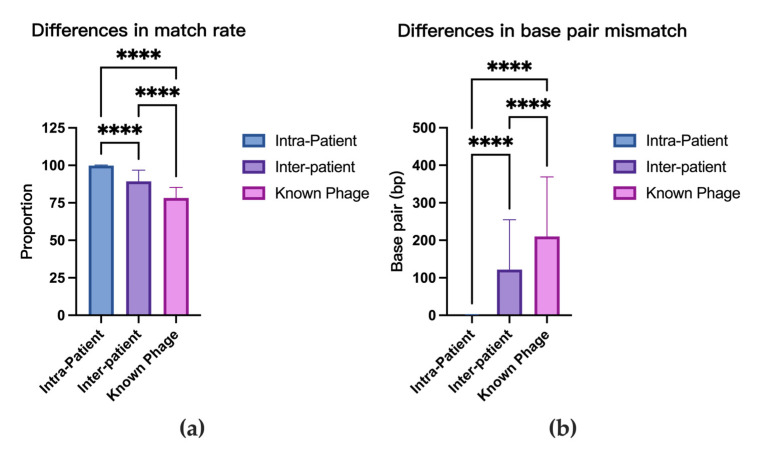
Differences in prophage sequence homology among intra-patient, inter-patient, and known mycobacteriophage comparisons. (**a**) Match rate differences. (**b**) Base pair mismatch differences. ****: *p* < 0.0001.

**Figure 7 microorganisms-13-02028-f007:**
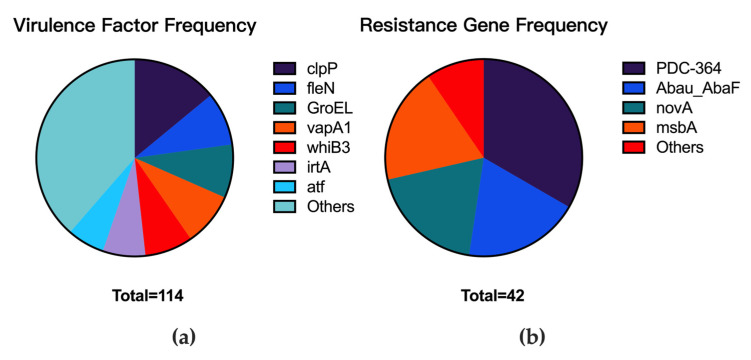
Frequency distribution of prophage-encoded virulence factors and resistance genes. (**a**) Frequency of virulence factor types detected within prophages. (**b**) Frequency of resistance gene types detected within prophages.

**Table 1 microorganisms-13-02028-t001:** Association between common genes encoded within prophages and patient prognosis. *: 0.01 ≤ *p* < 0.05.

	Favorable	Unfavorable	*p* Value
*clpP*	1 (16.7%)	4 (21.1%)	>0.9999
Undetected	5 (83.3%)	15 (78.9%)	
*fleN*	3 (50.0%)	4 (21.1%)	0.3442
Undetected	3 (50.0%)	15 (78.9%)	
*GroEL*	3 (50.0%)	4 (21.1%)	0.3442
Undetected	3 (50.0%)	15 (78.9%)	
*vapA1*	0 (0.0%)	2 (10.5%)	>0.9999
Undetected	6 (100.0%)	17 (89.5%)	
*whiB3*	4 (66.7%)	3 (15.8%)	**0.0324 ***
Undetected	2 (33.3%)	16 (84.2%)	
*irtA*	2 (33.3%)	4 (21.1%)	>0.9999
Undetected	4 (66.7%)	15 (78.9%)	
*atf*	2 (33.3%)	1 (5.3%)	0.1326
Undetected	4 (66.7%)	18 (94.7%)	
PDC-364	0 (0.0%)	3 (15.8%)	0.5539
Undetected	6 (100.0%)	16 (84.2%)	
*Abau_AbaF*	1 (16.7%)	4 (21.1%)	>0.9999
Undetected	5 (83.3%)	15 (78.9%)	
*novA*	1 (16.7%)	4 (21.1%)	>0.9999
Undetected	5 (83.3%)	15 (78.9%)	
*msbA*	1 (16.7%)	4 (21.1%)	>0.9999
Undetected	5 (83.3%)	15 (78.9%)	

## Data Availability

For bacteria genome data availability, all data have been deposited in OMIX at the China National Center for Bioinformation/Beijing Institute of Genomics, Chinese Academy of Sciences (https://ngdc.cncb.ac.cn/omix accessed on 1 October 2025, accession no. OMIX010879).

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
