# Peer review of "Characterization of Prophages and Their Genetic Cargo in Clinical M. abscessus Isolates"

_microorganisms, 2025, doi:10.3390/microorganisms13092028_

Round 1

Reviewer 1 Report

Comments and Suggestions for Authors

This study fills an important gap by describing prophages in clinical M. abscessus strains. The genomic data—especially the presence of virulence and antibiotic resistance genes—are new and valuable. However, the attempt to link prophage content to patient outcomes is weakened by the small and unbalanced patient group. The manuscript would be much stronger if it focused on the prophage genomics.

Key Issues in the Clinical Data Analysis

1. Small and Uneven Sample Size:
The study only includes 26 patients. Just 5 had a "good prognosis," which is too few for meaningful comparisons. This limits the ability to draw solid conclusions about how prophage content might relate to clinical outcomes.

2. Unclear Definitions:
The term “good prognosis” is defined as clinical improvement without bacterial clearance, which is confusing. The patient groups are also uneven—14 failed treatment, 5 improved, and only 1 cleared the infection. This makes group comparisons unreliable.

3. Repeated Strains from the Same Patient:
The study looks at 56 bacterial strains from 26 patients, with some patients contributing multiple strains. These strains are similar within each patient, so they shouldn’t be treated as independent data points. The analysis doesn’t correct for this.

4.- Were the strains representative of different M. abscessus subspecies (abscessusmassiliensebolletii) and infection types (pulmonary, skin/soft tissue)? Could lineage differences confound the results?

Conclusion on Prognosis Analysis:

Because of the small sample, unclear groupings, and overlapping data from the same patients, the study can't support solid links between prophage content and patient outcomes. The finding that better-outcome patients had more virulence genes is probably just due to chance.

Recommendation: Focus on Prophage Genomics

The real value of this study lies in its detailed look at prophages in M. abscessus (and sub species) —especially the discovery of new sequences and their genetic cargo. This is where the manuscript has the most impact.

Suggestions:

- Title: Focus on genomics (e.g., “Characterization of Prophages and Their Genetic Cargo in Clinical M. abscessus Isolates”)
- Abstract & Introduction: Emphasize the novel phage biology found in clinical strains, not the clinical outcomes. Present patient outcomes as context for strain collection, not as an analytical endpoint
- Discussion: Describe prophages as untapped sources of virulence and resistance genes in M. abscessus.

By shifting the focus, the manuscript will better highlight its strongest and most original findings.

Comments on the Quality of English Language

An example is the abstract. The abstract is poorly written and sometimes confusing. It contains grammatical errors, awkward phrasing, and imprecise language. For instance: “Little is known for lysogenic phages carried by M. abscessus” should be “Little is known about lysogenic phages carried by M. abscessus.”

“...nor their role played in diseases” should be “...nor about their role in disease.”

“...enriched virulence factors and drug resistance genes than known phages” is grammatically incorrect and unclear.

The sentence “The prophage detected in patients with good prognosis carried more virulence factors, and its meaning was unclear” contradicts expectations and is not explained.

Reviewer 2 Report

Comments and Suggestions for Authors

In this manuscript, authors described the research results on M. abscessus for the distribution of prophages carrying virulence factors and resistance genes, though clear association with clinical symptoms in patients was not found. The findings in this manuscript provide a scientific information for understanding of host-pathogen interaction.  This reviewer suggestis following points to revise this manuscript.

  1. Section 2.1. : How were the isolates identified as M. abscessus? Identification method should be written.
  2. Line 73: Expand "OADC".
  3. The rationale and the objective of this study should be clarified. Why did authors focus on prophage only? Are there virulence factor genes and resistance genes in the genome or plasmid? Why is the prophage more notable? 
